# Cardiac Implantable Electronic Miniaturized and Micro Devices

**DOI:** 10.3390/mi11100902

**Published:** 2020-09-29

**Authors:** Moshe Rav Acha, Elina Soifer, Tal Hasin

**Affiliations:** 1Jesselson Integrated Heart Center, Shaare Zedek Medical Center, Hebrew University, Jerusalem 910000, Israel; hasintal@gmail.com; 2Vectorious Medical Technologies, Tel Aviv 610000, Israel; elina@vectoriousmedtech.com

**Keywords:** implantable, cardiac, electrical, micro-device

## Abstract

Advancement in the miniaturization of high-density power sources, electronic circuits, and communication technologies enabled the construction of miniaturized electronic devices, implanted directly in the heart. These include pacing devices to prevent low heart rates or terminate heart rhythm abnormalities (‘arrhythmias’), long-term rhythm monitoring devices for arrhythmia detection in unexplained syncope cases, and heart failure (HF) hemodynamic monitoring devices, enabling the real-time monitoring of cardiac pressures to detect and alert for early fluid overload. These devices were shown to prevent HF hospitalizations and improve HF patients’ life quality. Pacing devices include permanent pacemakers (PPM) that maintain normal heart rates, defibrillators that are capable of fast detection and the termination of life-threatening arrhythmias, and cardiac re-synchronization devices that improve cardiac function and the survival of HF patients. Traditionally, these devices are implanted via the venous system (‘endovascular’) using conductors (‘endovascular leads/electrodes’) that connect the subcutaneous device battery to the appropriate cardiac chamber. These leads are a potential source of multiple problems, including lead-failure and systemic infection resulting from the lifelong exposure of these leads to bacteria within the venous system. One of the important cardiac innovations in the last decade was the development of a leadless PPM functioning without venous leads, thus circumventing most endovascular PPM-related problems. Leadless PPM’s consist of a single device, including a miniaturized power source, electronic chips, and fixating mechanism, directly implanted into the cardiac muscle. Only rare device-related problems and almost no systemic infections occur with these devices. Current leadless PPM’s sense and pace only the ventricle. However, a novel leadless device that is capable of sensing both atrium and ventricle was recently FDA approved and miniaturized devices that are designed to synchronize right and left ventricles, using novel intra-body inner-device communication technologies, are under final experiments. This review will cover these novel implantable miniaturized cardiac devices and the basic algorithms and technologies that underlie their development. Advancement in the miniaturization of high-density power sources, electronic circuits, and communication technologies enabled the construction of miniaturized electronic devices, implanted directly in the heart. These include pacing devices to prevent low heart rates or terminate heart rhythm abnormalities (‘arrhythmias’), long-term rhythm monitoring devices for arrhythmia detection in unexplained syncope cases, and heart failure (HF) hemodynamic monitoring devices, enabling the real-time monitoring of cardiac pressures to detect and alert early fluid overload. These devices were shown to prevent HF hospitalizations and improve HF patients’ life quality. Pacing devices include permanent pacemakers (PPM) that maintain normal heart rates, defibrillators that are capable of fast detection and termination of life-threatening arrhythmias, and cardiac re-synchronization devices that improve cardiac function and survival of HF patients. Traditionally, these devices are implanted via the venous system (‘endovascular’) using conductors (‘endovascular leads/electrodes’) that connect the subcutaneous device battery to the appropriate cardiac chamber. These leads are a potential source of multiple problems, including lead-failure and systemic infection that result from the lifelong exposure of these leads to bacteria within the venous system. The development of a leadless PPM functioning without venous leads was one of the important cardiac innovations in the last decade, thus circumventing most endovascular PPM-related problems. Leadless PPM’s consist of a single device, including a miniaturized power source, electronic chips, and fixating mechanism, implanted directly into the cardiac muscle. Only rare device-related problems and almost no systemic infections occur with these devices. Current leadless PPM’s sense and pace only the ventricle. However, a novel leadless device that is capable of sensing both atrium and ventricle was recently FDA approved and miniaturized devices designed to synchronize right and left ventricles, using novel intra-body inner-device communication technologies, are under final experiments. This review will cover these novel implantable miniaturized cardiac devices and the basic algorithms and technologies that underlie their development.

## 1. Introduction

Advancement in the miniaturization of high-density power sources, electronic circuits, and communication technologies enabled the construction of miniaturized electronic devices, implanted directly in the heart. These include pacing devices to prevent low heart rates or terminate rapid heart rhythm abnormalities (‘tachyarrhythmias’), long-term rhythm monitoring devices for arrhythmia detection in unexplained syncope cases, and heart failure (HF) hemodynamic monitoring devices, enabling the real-time monitoring of cardiac pressures to detect and alert for early fluid overload. These devices were shown to prevent HF hospitalizations and improve HF patients’ life quality. Pacing devices include permanent pacemakers (PPM) that maintain normal heart rates, defibrillators that are capable of fast detection and the termination of life-threatening arrhythmias, and cardiac resynchronization devices that may improve cardiac function and survival in appropriately selected HF patients. Traditionally, these devices are implanted via the venous system (‘endovascular’), while using conductors (‘endovascular leads/electrodes’) that connect the subcutaneous device battery to the appropriate cardiac chamber. These leads are a potential source of multiple problems, including lead-failure and systemic infection that result from the lifelong exposure to bacteria within the venous system. One of the important cardiac innovations in the last decade was the development of a leadless PPM functioning without venous leads, thus circumventing most endovascular PPM-related problems. Leadless PPMs consist of a single device, including a miniaturized power source, electronic chips, and fixating mechanism, implanted directly into the cardiac muscle. Only rare device-related problems and almost no systemic infections occur with these devices. Current leadless PPMs sense and pace only the ventricle. However, a novel leadless device capable of sensing both atrium and ventricle was recently FDA approved and miniaturized devices that were designed to synchronize right and left ventricles, using novel intra-body inner-device communication technologies, are under final experiments. This review will cover these novel implantable miniaturized cardiac devices and the basic algorithms and technologies underlying their development.

## 2. Conventional Pacing Devices

Cardiac implantable electronic devices (CIED) have become a pivot treatment for various cardiac patients, including: (a) permanent pacemakers (PPM) aimed for pacing the heart at programmed rates, for patients with symptomatic low heart rates (‘bradycardia’). (b) Implantable cardiac defibrillators (ICD), which can detect and terminate life-threatening fast ventricular arrhythmias via anti-tachycardia pacing or high-energy shocks (‘direct current cardioversion/defibrillation’). These ICDs were shown to improve the survival of HF patients with impaired cardiac function, who are at increased risk for sudden arrhythmic death. ICD is also recommended for patients surviving prior arrhythmic cardiac death, considered to be high risk for recurrent ventricular arrhythmias [1,2,3,4,5]. (c) Cardiac resynchronization therapy devices (CRT) for HF patients who have both severely reduced cardiac function and an impaired electrical conduction, which resulted in asynchronous cardiac contraction. Current CRT device function is based on achieving synchronous contraction of both right and left ventricles, which results in improved global mechanical performance of the heart. CRTs were shown to attenuate HF symptoms and they may improve survival in HF patients [6,7]. All of the above devices consist of a battery, programmable electrical chips (consisting of electrical timing circuits and output drivers), and conductors (also named ‘leads’ or ‘electrodes’). The devices are usually implanted below the skin (‘subcutaneously’) in the chest area, and a set of leads are advanced from the device pocket, via the central venous system, into various parts of the heart (Figure 1). Conventional PPM leads are composed of a metal conductor (‘coil’) enclosed by insulation layer, which minimizes the current leak and electromagnetic interference from the outside world (Figure 2). In contrast with old unipolar leads, where the lead tip serves as the cathode and the device battery as anode, most modern leads are bipolar leads, where the lead tip serves as cathode and a proximal “ring” serves as the anode. The lead conductors are arranged in coaxial or coradial designs (Figure 2) [8,9]. *Coaxial* leads have an inner conductor that extends down the length of the lead to the tip electrode, the cathode, arranged in a coil configuration with a central lumen to allow for the passage of a stylet at implantation. This coil is covered by a cylindrical length of inner insulation, which, in turn, is wrapped by another coil conductor that also runs down the lead to the ring electrode, the anode. A second outer insulation layer protects the ring conductor from the outside environment (Figure 2B). In coradial leads, a single coil extends down the length of the lead and it consists of two parallel, alternating conductor strands, one of which connects to the cathode and the other to the anode. Each conductor strand is individually coated with an inner insulation layer and the single two-component coil is surrounded by another outer insulation layer (Figure 2). Although coaxial design leads have been the standard for many years, the resulting bulk and stiffness of this four-layer design encouraged the development of the coradial design, which is less bulky (~1.7 mm) and more flexible. The outer insulation in both designs is made of polyurethane or silicone. Polyurethane is a synthetic segmented polymer with high tensile strength and resistance to mechanical abrasion. Thus, a thin layer of insulation can be used in order to cover the lead conductors, enabling low diameter leads. However, polyurethane leads are stiff and not fully biostable, being subject to in vivo biological degradation, due to environmental *stress cracking* and *metal ion oxidation*, resulting in insulation defects. Silicone, in contrast, is more flexible and biostable over extended periods. Its main disadvantage relates to its lower tensile strength, thus rendering it susceptible to abrasion and tears. Therefore, a thicker insulation layer must be used to maintain lead reliability, which increases lead bulk [10,11].

MP-35N, an alloy of nickel, cobalt, chromium, and molybdenum, is the basic material that is used for most modern conductors. The main advantage of MP-35N is its high strength and resistance to corrosion [8]. Its main disadvantage is its high electrical resistance, but this has been overcome with the development of composite-wire conductors that incorporate low-resistance metals, such as silver (thus named silver cored alloy wires) [8]. In most pacing leads, these materials are generally incorporated into a *drawn filled tube* composite-wire conductor strand consisting of a thick strong body of MP-35N filled with a central core of softer low-resistance silver metal, often encased in a further outer shell of platinum alloy. An alternative conductor design is the *drawn brazed strand* design, consisting of few strands of high-resistance MP-35N material tightly molded over a central silver strand. Thus, the inner low-resistance metal is forced between and around the strong outer strands. The functional purpose of these designs is to create conductors that are both resistant to corrosion and have low electrical resistance (Figure 2).

Importantly, all of these devices are defined as endovascular devices, due to the uniform need to use long leads throughout the venous system. These leads are fixed to the appropriate chamber by small tines (passive leads), or via a screw into the heart muscle (Figure 1). The main caveat of all above devices results from having long metallic leads within the venous system for many years. This may cause various complications throughout the life-time of the device [12]. These include early complications during device endovascular implant procedure, such as perforation of the cardiac chamber by the fixation mechanism [13,14,15,16] or lung injury (‘pneumothorax’) [17,18,19,20] caused by puncture of the lungs while trying to access the central vein. Late complications include lead failure due to insulation defects or fracture (see insulation properties discussed above), early and late lead dislodgements (before or after six weeks post implant, respectively) necessitating re-operation for lead revision [20,21,22,23,24,25], and a potentially lethal systemic infection (‘endocarditis’) that is caused by bacteria adhering to these endovascular metal electrodes potentially spreading infectious emboli from the heart to all body organs [26,27,28,29,30,31]. Indeed, device endocarditis is one of the major complications associated with conventional endovascular CIEDs, and it is the main reasons for minimizing their use when not obligatory [12,30,31]. The development of leadless PPMs, as an option to circumvent most of the above endovascular-related complications, was a major breakthrough regarding CIED. The current functional leadless PPM and long-term arrhythmia monitoring devices, as well as recent innovations regarding novel defibrillation and leadless CRT devices, will be discussed in the following paragraphs.

## 3. Leadless PPM

The development of leadless PPM became reality due to advances in electrical engineering, material design, and computer science. Most notably, advances in large-scale integration circuitry have resulted in highly compact electric circuit design with low-power consumption, which enabled the construction of implantable miniaturized leadless PPMs. This modern circuit design, already used in most modern conventional PPM, is based on complementary metallic oxide semiconductor (CMOS) technology that is extremely compact and operates at low energy cost (Figure 1, bottom). Leadless PPM consist of a pulse generator that is attached to the device electrical circuits, a sensing/pacing micro-electrode, which is an inherent part of the device and not a separate long conductor, as in conventional PPM, and a fixation mechanism. All of these device contents are hermetically enclosed and contained within a single miniaturized device (25–40 mm length, volume of 0.8–1 cm^3^ weighting ~2 g), which is implanted as one piece directly into the appropriate cardiac chamber (Figure 3). Notably, the volume of these devices is ~one-tenth of that of a conventional single chamber PPM resulting in ~93% reduction in the amount of non-biological material within the body. Moreover, with time, these small leadless devices may become encapsulated within the cardiac wall, so that no metal is exposed to the bloodstream. Initially, two leadless PPM were on the market (Nanostim, Saint Jude Medical, Little Canada, MIN, USA [32,33]; Micra, Medtronic, Minneapolis, MIN, USA [34,35]).

The leadless PPMs differ in size, fixation mechanism, communication, rate response mechanism, and power source. Micra is smaller as compared with Nanostim (0.8 cm^3^ vs. 1 cm^3^). Micra’s fixation is based on nitinol tines at its distal end, which are pushed into to myocardium, while Nanostim’s fixation is based on an active helix that is screwed into the cardiac muscle. The communication with remote programmer ‘telemetry’ is based on radiofrequency signals for Micra versus conductive communication through skin electrodes for Nanostim [34]. Both of the devices have rate-response features, i.e a specific algorithm and technology designed to detect patient movements and increase the pacing rate accordingly, mimicking the physiological heart rate increase during exercise. Micra’s rate-response is based on a three-axis accelerometer, allowing for rate-responsiveness to change in magnitude during activities as climbing stairs (where there are mainly vertical and less horizontal movement changes as compared with running), while Nanostim’s rate-response activity is based on blood temperature sensor, sensing increased temperature during patient activity (Table 1). A lithium silver oxide/carbon monofluoride battery powers Micra, and high-density lithium carbon monofluoride battery powers Nanostim. These light-weight batteries provide high current densities that support onboard pacing without voltage dips, which were previously reported with other batteries [36]. Both of the devices use a special delivery system based on a tethering mechanism to maintain connection between the delivery catheter and the device during positioning maneuvers until a final location is established, where the device is deployed and disengaged from the delivery catheter.

Apart of the technological advances that are described above, a crucial step in leadless PPM development was achieved by reducing their energy requirements, which enabled the use of a miniaturized power source. These lower energy requirements result from: (1) construction of efficient low-power electrical circuits; (2) absence of long high-impedance leads, saving energy needed for current conduction along the long conventional leads; (3) automatic capture management algorithm (measuring pacing output threshold to achieve myocardial capture), processed on an hourly basis to ensure minimal but safe pacing outputs; and, (4) optimized battery longevity by setting the nominal pulse width duration to chronaxie (0.24 ms) [37]. The presence of high density batteries along with the reduced energy requirements of these devices enable an estimated longevity of 10 and 14.7 years for Micra and Nanostim, respectively; assuming 1.5 V/0.24 ms pacing output set to a lower pacing rate of 60 beats per minute with a pacing impedance of 500 ohm and 100% pacing. The device longevity difference between these devices results from Micra’s smaller battery and its radiofrequency-based telemetry, requiring greater current drain as compared with Nanostim’s conductive telemetry [34]. Importantly, a prerequisite for development of miniaturized leadless PPM derives from the basic fact that cardiac pacing as a whole requires a little amount of current. On the contrary, cardiac defibrillation depends on high-voltage shocking coils, which prevents the development of miniaturized ICD.

The implantation of leadless PPMs was shown to be successful in the vast majority of cases (~95%) and free from serious complications [32,33,34,35,36]. The one year results of the LEADLESS trial (utilizing Nanostim) revealed that: (1) device performance, including pacing threshold, impedance, and sensing remained stable, without over/under-sensing issues; (2) no complications were associated with the device beyond the index implantation procedure; specifically there were no device embolization seen, confirming an adequate and reliable fixating mechanism; (3) no premature battery depletions; and, (4) adequate rate response noticed in those cases in which it was activated [38]. Similar results were found for Micra [39].

Despite this early success of leadless PPM, the clinical use of Nanostim was halted by the end of 2017 due to two safety issues: (1) premature battery depletion that was initially attributed to “lithium clusters” bridging the cathode and anode, causing electrical short-circuiting. A high percent of device malfunction attributed to early battery depletion was reported in a single center study [40]. Notably, few of the malfunctioning devices in that study were retrieved, revealing reduced electrolytes within the lithium carbon monofluoride battery, which resulted in a high internal battery resistance. This impacts the current available to power the device electronics and it results in a loss of device functionality [40]; and, (2) detachment of the docking button, which is a 3.6 mm component connected to the end of the device, designed to allow for device retrieval by connecting the device with the retrieval catheter [41]. As of today, a new Nanostim version, overcoming the above issues, is under final developmental stages and should soon be available.

The main limitation of leadless PPM results from the fact that they could pace and sense the right ventricle (RV) only without Atrio-Ventricular (AV) synchronous pacing, which is considered to be beneficial in most patients. Thus, their use is restricted to symptomatic bradycardia patients without the need for atrial sensing or pacing as patients with slow chronic atrial fibrillation, some patients with alternating fast atrial fibrillation and sinus bradycardia where no significant atrial pacing is needed, and elderly sedentary patients with multiple co-morbidities in whom ventricular pacing is sufficient. A second limitation results from device encapsulation into the myocardial tissue, occurring as soon as one year post-implantation, as described in a few autopsy studies [42,43]. This may result in difficulty in retrieving these devices after that period, although successful device extractions have still been reported even after one year post implantation [44]. This same limitation might be a great advantage of these devices, as it may explain the fact that there are almost no infections that are associated with these devices [45,46]. This is probably related to their miniaturized volume, continuous movement with the heart (preventing bacteria from adhering), and their incorporation within the cardiac wall, whereby there is no longer exposure of a metallic material to the bloodstream. Due to their inherent leadless nature, circumventing most lead-related problems with almost absence of device-associated systemic infections, the need for retrieval of these devices was shown to be ~80% lower than conventional PPMs [39,47]. On the whole, there is no doubt that these leadless PPMs are associated with less complications in general and, specifically, with less systemic infections.

Recently, effort was made to expand current leadless PPM use by developing a novel atrial sensing algorithm for enabling AV synchronous pacing, which is the “physiological” optimal pacing method for most patients suffering from low heart rates. This algorithm is based on a complex analysis of Micra’s accelerometer signals, distinguishing intracardiac signals that are related to atrial contractions from those related to ventricular movement [48]. This novel algorithm is based on complex analysis of mechanical-based signals while using advanced filtering and rate-smoothing algorithms, by which the timing of atrial contraction is decoded, translating this mechanical information to the assumed timing of the electrical atrial P wave. Thereafter, the timing of ventricular pacing occurs after a programmable duration period, mimicking the physiological electrical activity that is initiated by the sinus node within the right atrium conducting to the ventricles via the conduction system with a physiological delay within the atrioventricular node (AVN). This AVN delay is crucial in enabling enough time for complete blood transfer from the atria into the ventricles before these start to contract (since, during ventricular contraction, the atrioventricular valves close and do not allow further transfer of blood from the atria). By this algorithm, leadless ventricular PPM could extend their function from ventricular pacing and sensing only (VVI) system to a ventricular pacing with both an atrial and ventricular sensing (VDD) system. Such a VDD system could sense atrial contraction and coordinate ventricular pacing within a programmable AV delay period, in order to enable physiologic AV synchronous pacing. As of today, few feasibility trials were conducted to test this algorithm, revealing 87% successful AV synchronous pacing [48]. An updated version of a VDD Micra has just recently been released for clinical practice.

## 4. Modular Subcutaneous ICD and Leadless PPM

Basically, PPMs are used to treat slow heart rhythms, while fast potentially life-threatening ventricular arrhythmias, which could result in sudden death if untreated, are terminated by electrical shocks or anti-tachycardia pacing (ATP). Current implantable cardiac defibrillators (ICD) are utilized for the rapid detection of fast ventricular arrhythmias and their termination by automatic DC shocks or ATP. Conventional ICDs are composed of a high-energy battery, trans-venous leads containing coils (capacitors), which are charged to high voltage, enabling the delivery of high-energy DC shocks through the leads into the heart muscle when needed. Similar to PPMs, these devices are prone to all endovascular-related complications. As noted earlier, cardiac defibrillation depends on high-voltage shocking coils, which limits (for the time being) the development of miniaturized ICDs due to these high-energy requirements that are beyond the capability of current miniaturized batteries. Nevertheless, a significant reduction of ICD volume occurred over the years that was mainly due to a reduction in its capacitor size.

A subcutaneous ICD (S-ICD) was developed in order to circumvent endovascular-related complications (EMBLEM-MRI; Boston Scientific, Marlborough, MA, USA; Figure 4). This device uses long subcutaneous lead with a shocking coil that is placed above the heart area, which enables a high-voltage DC shock to affect the heart rhythm (similar to an external defibrillator shock) [49]. However, these devices have no pacing capability, since both the device itself and its lead are located subcutaneously, without contact with the cardiac muscle. Accordingly, a novel modular device, which utilizes both S-ICD and a miniaturized leadless PPM (EMPOWER™ leadless cardiac pacemaker), was developed to enable both cardiac pacing and defibrillation, via inter-device conductive radio-frequency near-field communication (investigational device, Boston Scientific) [50,51]. The modular device can also terminate fast arrhythmia via ATP, given by the leadless PPM in response to S-ICD ‘request’, due to its pacing capability. The intra-body radio-frequency inter-device communication is based on low voltage 25 KHz alternating current pulses, sent in proprietary pattern from the S-ICD shocking coil (functioning as a transmitting electrode) to the S-ICD battery (functioning as a receiving electrode), which creates a communication vector between them using the body tissue as a conductor. A dipole electrode within the leadless PPM, which is positioned in the right ventricle within the path of the communication vector, can sense these signals [51,52]. Although such communication is dependent on the distance between the transmitting and receiving electrodes, body tissue (‘conductor’) impedance, and the orientation angle of the leadless PPM dipole electrode relative to the communication vector, initial studies have shown such intra-body communication to be safe and effective [51,52]. Notably, subcutaneous ICD performance was found to be noninferior to transvenous ICD in a recent large multicenter study [53].

## 5. Multi-Component Leadless Cardiac Resynchronization Pacing System

The synchronization of right and left ventricular (LV) function was shown to improve cardiac function, prevent hospitalizations, and prolong survival in a selected subpopulation of HF patients with an electrical conduction disease, having asynchronized ventricular contraction. Conventional cardiac resynchronization therapy (CRT) devices are based on multiple transvenous leads, implanted within the right atrium, right ventricle, and a cardiac venous branch of the coronary sinus, which, due to its proximity to the LV, enables LV pacing. These leads pace the ventricles simultaneously, resulting in a synchronized improved cardiac contraction. Multi-component leadless PPM implanted directly within both RV and LV, enabling their synchronous pacing, are under development [54]. One of these first attempts is the WISE-CRT system utilizing novel ultrasound-based leadless cardiac stimulation [55]. This system is based on a previously implanted conventional ICD devices with their endovascular RV leads along with a new miniaturized receiver/pacing lead directly implanted into the LV, and a subcutaneous pulse generator (comprised of a battery and a transmitter) implanted just above the LV lead. Cardiac resynchronization is achieved by the subcutaneous pulse generator that detects RV pacing signals and nearly simultaneously generates and transmits ultrasonic acoustic energy signals to the receiver/pacing LV electrode that converts these signals to electrical pacing pulses [55]. A re-designed similar SELECT-LV system was developed due to serious implant procedure-related complications, mainly cardiac perforations, which enabled safe implantation with adequate cardiac resynchronization and improved cardiac function [56]. This system still has multiple caveats, including: (1) the need for an endovascular ICD lead; (2) dependence on acoustic window, necessitating alignment of the LV electrode and the subcutaneous transmitter that is not always feasible; (3) long-term use of ultrasound energy which may have unintended adverse effects on myocardial tissue and might be exposed to external interference; and, (4) potential emboli of the LV electrode resulting in stroke. An international registry evaluating the efficacy and safety of the WISE-CRT pacing system was just recently published [57]. The registry included 90 patients who either failed or could not undergo a conventional trans-venous CRT implant, due to no venous access or anatomic constrains. The registry revealed high implant success rate (94%), but with a very high six-month complication rate (24%), including: two patients with acute cardiac perforation, two patients with acute lung injury, four deaths of whom three were considered procedure-related, one stroke, three patients with infection, and four patients with arterial bleeding [57].

Recently, a totally leadless CRT system was developed, which was based on three independent leadless PPM’s implanted directly into the right atrium, right ventricle, and left ventricle [58]. Each of those leadless PPM’s consists of a pacing and a communication nodule, which have been implemented with a microcontroller and field-programmable gate array board. Each PPM electrode is used for both transmitting and for receiving signals, acting as communication interface electrode. The device’s synchronous activation is based on a novel ultra-low power Conduction Intra-Cardiac Communication (CIC) method. This novel communication method allows for multidirectional communication between the system independent devices without disturbing the biologic intra-cardiac electrical signals used for physiologic cardiac pulse conduction, while using minimal energy consumption that is suitable for long-term continuous action supported by a miniaturized device battery. Notably, conventional radio-frequency communication (traditionally used for transient communication between implanted deices and remote programmers-‘telemetry’) has high power absorption within the human body, which results in high-energy consumption. Radio-frequency communication is not suitable for continuous long-term intra-body device communication due to this high-energy consumption and susceptibility to external electromagnetic interference.

The novel CIC method is based on several very-high-frequency small alternating current pulses that were transmitted and received by the devices, whereby communication data are encoded in the time shifts between pulses. The alternating current communication signal propagates almost at the speed of light and allows for the transition of several bits of data per pulse. Using very high frequency signals has major advantages, which are prerequisite for long-term intra-body communication: (1) low signal absorption and attenuation within cardiac tissue, resulting in highly efficient energy communication; (2) does not interfere with intrinsic cardiac electrical activity used to both conduct biological electrical pulses throughout the cardiomyocytes and depolarize the cardiomyocytes’ membrane to enable cardiac contraction (known as ‘excitation-contraction coupling’), thus preventing inappropriate pacing with its potentially deleterious consequences; and, (3) low risk for external electromagnetic interference. A communication frequency of 1 MHz was found to be optimal, causing no cardiac intrinsic electrical interference and achieving a minimal (0.3 µW) power requirement. Notably, each device pacing action is triggered by a CIC message sent from another device containing sender and receiver address (which cardiac device chamber sends/receives data). As the CIC message travels through the heart, the received signal undergoes some attenuation and it is also affected by noise. Accordingly, the communication nodule within each device amplifies and filters the incoming signal in order to obtain the reconstructed messages, which can then trigger a pacing stimulus. The above experimental system was shown to successfully resynchronize cardiac function in porcine hearts and only recently tried in humans [58,59,60]. Interestingly, recent successful implantation of a leadless CRT system [61], as well as a combined implantation of leadless CRT system along with S-ICD system [62], were recently reported in two patients, revealing the feasibility of a combined implant of these two separate systems without electrical interference [62].

Importantly, one should be aware of the embolic risk that is associated with any metal device implanted within the LV or left atrium (LA) (see below, HF monitoring devices). This may result from the detachment of the LV/LA device itself or from thrombus formation on these metal-based devices, which, due to the vigorous cardiac contractions and high pressure blood stream, may embolize into the aorta, causing stroke or systemic embolism. A continuous lifetime anticoagulation treatment is necessary to prevent a thrombus formation on these LV devices, although such treatment may lead to bleeding complications and, at times, needs to be stopped (before surgery or other invasive procedures). Trying to prevent the embolic detachment of LV device itself is a challenge, since it requires a firm screw or other fixation mechanism to firmly attach to the LV wall. However, this firm attachment may cause too much pressure on the LV wall, resulting in cardiac perforation, which is the second complication that is associated with these devices. Indeed, the main challenge at present is to find the appropriate fixation mechanism to ensure enough strength to prevent dislodgment, but without the risk of cardiac wall perforation. Notably, device emboli to lower extremities and a cerebral stoke occurred among 2/35 patients that were implanted with the SELECT-LV system [56]. Additionally, in a British registry of 68 HF patients who failed conventional CRT devices and were implanted with endocardial LV leads, 6% suffered a stroke over a 20 month follow-up period [63].

Noteworthy, a few novel ‘physiologic’ pacing paradigms were recently developed, as an alternative to conventional CRT pacing. These include His bundle and left bundle branch (LBBB) pacing, in which a conventional PPM device is connected to a special lead positioned in the vicinity of the His bundle or LBBB one, respectively [64,65,66,67,68,69,70,71,72]. This is in contrast with the classic pacing paradigm, in which the pacing lead is positioned close to the RV apex (Figure 5). The idea behind both of these novel pacing modalities is the ability to capture and utilize the physiologic conduction system. This system is composed of specialized bundles (His, left, and right bundle branches, and the distal spreading Purkinje fibers) that are capable of fast and reliable electrical pulse delivery to all cardiomyocytes, underlying normal synchronous electrical pulse propagation within the heart (Figure 5). By pacing the physiological conductive system, one could potentially achieve an ideal synchronization of both RV and LV. This new pacing paradigm is in contrast to the classic pacing paradigm, where RV pacing lead is usually located at the RV apex, which is outside the physiological conduction system. Accordingly, classic RV lead pacing results in an asynchronous contraction of the RV and LV, which necessitates the use of another LV pacing electrode, whereby the activation of both simultaneously may restore the synchronization of RV and LV (the concept underlying CRT). However, the use of the novel His and LBBB pacing paradigms, utilizing the physiological conduction system to achieve perfect synchronization of all cardiomyocytes within both RV and LV, obviate the need for multiple leads within both RV and LV for synchronization. Initial experience with both of these pacing systems reveal promising results [66,67,68,69,70,71,72]. Nevertheless, to the best of our knowledge, miniaturized devices that are specifically designed for His or LBBB pacing are not currently available.

## 6. Implantable Long-Term Electrocardiographic Monitoring Device

The development of a miniaturized implantable loop recorder (ILR) for long-term monitoring of arrhythmias made a revolution in the evaluation of patients suffering from recurrent fainting (‘syncope’) episodes [73,74], strokes (related to atrial flutter and fibrillation) [75,76,77,78,79] as well as risk stratification of various arrhythmogenic syndromes [80,81,82]. For years, the clinical evaluation of these patients included multiple 24 h ECG recordings (‘Holter’) to search for possible arrhythmias that might underlie syncope or stroke. The finding of specific arrhythmias would mandate a specific treatment to prevent further episodes. The 24 h Holters had a low yield for detecting arrhythmias, which are usually infrequent and of paroxysmal nature. The development of ILR enabled a continuous long-term (up to three years) recording and automatic identification of arrhythmias, which results in a dramatic increase in its diagnostic yield. ILRs are composed of a miniaturized battery and sensor electrode, which are implanted within minutes subcutaneously in the chest above the heart, enabling long-term high quality cardiac rhythm monitoring [83,84]. Once an arrhythmia is detected, it is recorded by the ILR, which then sends an automatic alarm to the patient’s clinic via internet connection, in order to facilitate diagnosis and the selection of additional appropriate therapy. After completing their mission (arrhythmia detection) or once their battery life is over, the ILR could be explanted via a simple procedure with a tiny remnant scar. As of today, there are thousands of syncope and stroke patients who owe their accurate and, at times, life-saving diagnosis to these ILRs, after multiple previous 24 h Holters failed to diagnose their arrhythmias [73,74,78,85].

## 7. Implantable Devices to Monitor Heart Failure (HF) Hemodynamics

HF is a chronic syndrome that is caused by inability of the heart to pump sufficient blood to the body tissues along with elevated cardiac filling pressures. These translate to fluid overload in the lungs and tissue hypoperfusion, which will eventually lead to pulmonary congestion and end-organ dysfunction, correspondingly. The clinical consequences of these changes include symptoms, such as shortness of breath due to pulmonary congestion, peripheral edema due to inadequate pumping of blood from the extremities, general weakness, attenuated exercise tolerance, and multiorgan dysfunction due to inadequate blood supply. Although some patients remain stable for years, most will suffer from exacerbations that may lead to hospitalizations, disability, and eventually death. The timely detection of exacerbations before overt clinical manifestations ensue will trigger identification of reversible causes and appropriate treatment to prevent hospitalization and further deterioration. To this end, several intracardiac microdevices have been developed and put into clinical use. The leading concept behind these intra-cardiac devices for HF monitoring is that cardiac filling pressure increases days to weeks before overt clinical exacerbation occurs [86,87]. Several technologies were developed over the past decade in order to accurately measure pressure at different sites along the cardiac circulation and transmit this information to a care facility, where care-givers can appropriately react by changing the treatment dose and strategy [87,88].

The first of these devices that was clinically tested was the Chronicle implantable continuous hemodynamic monitor (Medtronic, Inc., Minneapolis, MIN, USA). This device resembles a PPM, with a programmable component that processes and stores information that is similar in appearance to the pulse generator of a conventional PPM, and a transvenous lead that has a sensor incorporated near its tip, which enables continuous intra-cardiac pressure measurements. The lead tip was inserted into the RV near the outflow tract and collected RV systolic and diastolic pressures. The leading clinical trial (COMPASS-HF), randomized HF patients to HF treatment with and without hemodynamic monitoring, revealing no significant change in HF-related adverse events apart from delaying the time to first HF-related hospitalization [89].

A significant success with hemodynamic monitoring in HF was first demonstrated while using the CardioMEMS pulmonary artery (PA) pressure monitoring system (Abbott, Sylmar, CA, USA). This novel wireless device with a battery free technology is powered and interrogated via an external antenna, using electromagnetic coupling. The device is implanted into a branch of the PA where pressure to its sensor causes the deflection of the pressure-sensitive surface, causing a shift in the resonant frequency. The device external antenna is usually embedded within a pillow on which the patient lies, so pressure transmissions occur while the patient is resting. Clinical utilization was tested in the CHAMPION trial that randomized 550 HF patients and demonstrated a significant reduction in the rate of HF hospitalizations [90]. This pivotal trial showed, for the first time, the ability of an implanted electronic device to transmit accurate real-time hemodynamic changes occurring within the hearts of HF patients, heralding symptomatic HF exacerbations. Given these transmitted subclinical changes, a physician can instruct the patients to change their medications in order to counteract these hemodynamic changes, for example, by increasing diuretic medications to facilitate fluid removal and prevent full blown clinical HF exacerbation, which would otherwise necessitate prolonged hospitalization. Based on this trial as well its follow up trial, the U.S. Food and Drug Administration approved the CardioMEMS HF System for use in patients with advanced HF, who had at least one HF hospitalization in the previous year before implantation [90,91,92].

While pulmonary pressure monitoring was found to be successful, the direct measurement of left atrial pressure (corresponding to LV filling pressure) is thought to have further clinical advantages for HF monitoring. Left atrial pressure (LAP) is less influenced by pulmonary disease and pulmonary vascular remodeling, and it may be a better marker of response to HF medications. Furthermore, animal studies demonstrated a strong correlation between increases in LAP and pulmonary congestion. The utility of direct LAP monitoring was first evaluated using the HeartPOD system (Abbott, formerly St. Jude Medical/Savacor, Inc., Saint Paul, MN, USA). This system includes an implantable sensor lead coupled to a subcutaneous antenna coil, a patient advisory module, and remote clinician access via secure computer-based data management. The tip of the sensor system lead was transvenously implanted into the left atrium via the atrial septum. The implant was powered and interrogated through the skin by wireless transmissions from the patient advisory module. The LAPTOP-HF trial was intended to evaluate the utility of this system, but it was stopped early due to a perceived excess of implant-related complications [93].

Newer devices to measure and transmit LAP with promising initial results include the V-LAP system (Vectorious Medical Technologies, Tel Aviv, Israel) [94]. This is a digital miniature (<18 mm length, 3.9 mm diameter) wireless and leadless sensor (Figure 6A), which is implanted permanently in the interatrial septum via a minimally invasive procedure. The device incorporates an advanced micro application-specific integrated circuit (ASIC) technology, allowing for onboard drift compensation. The system also includes an easily portable external wearable device (Figure 6B), which remotely powers and interrogates the implant via radio frequency bi-directional communication upon activation, and then transmits the collected LAP data, including a high-resolution LAP waveform (Figure 6C), to a secure cloud-based database. The data are accessible for clinician review via a designated web application. The first in-man VECTOR-HF trial is currently recruiting patients.

Notably, among HF patients implanted by an ICD or CRT devices, one could use these intracardiac devices to promote continuous monitoring of various hemodynamic parameters to facilitate diagnosis of HF exacerbations [95,96,97,98,99,100,101,102,103]. Indeed, various algorithms derives from these intracardiac devices were shown to promote subclinical HF exacerbations [100,101,102,103].

## 8. Conclusions

Major technological advances, including large-scale integration circuitry using complementary metallic oxide semiconductor (CMOS)-based electrical chips or advanced microelectronic chips (ASIC), which result in highly compact electric circuits with low-power consumption, underlie the development of various implantable miniaturized devices used for pacing and real-time monitoring of cardiac patients. Thereafter, use of novel communication technologies enable fast and reliable intracardiac inter-device communication with minimal interference from cardiac motion and external signals. These miniaturized leadless devices, which are directly implanted into the heart, overcome most endovascular-related complications that are associated with conventional pacing devices. Although many of these devices are still not micro-scale, current significant size reduction, along with novel medical engineering approaches, suggest real micro-scale devices are forthcoming.

As of today, single chamber (usually RV) leadless pacing systems are widely used and multiple leadless pacing systems, such as the leadless CRT, are under clinical trials. Initial studies on leadless CRT pacing systems reveal good implant efficacy, but with high complication rates. Should these multiple leadless device systems prove to be safe and feasible, this would enable the wide use of leadless pacing systems for all cardiac pacing and resynchronization purposes. Last and not least, new technology is needed in order to develop miniaturized high-energy power sources to enable the development of miniaturized ICDs. Until then, leadless pacing systems would need to interact with endovascular conventional ICD or subcutaneous ICD.

Regarding real-time hemodynamic monitoring of HF patients, various implanted devices were developed in order to measure cardiac pressure and utilize it for clinical practice (Table 2). The only one currently available for clinical practice is the CardioMEMS, which captures and transmits PA pressure. Technical solutions have evolved from pacemaker-like to miniature wirelessly charged devices. Measurement sites evolved from the RV to the PA and eventually the LA, which results in direct measurements of LV filling pressures. With the new reality enforced by COVID-19 pandemic, the importance of these sensor-type devices has been further enforced. The advantage of reporting devices is highlighted, as many HF patients are self-segregated or refrain from seeking medical advice. Future development of new devices detecting and transmitting other signals besides pressure, such as saturation, blood viscosity, and temperature, will be evaluated for their usefulness in monitoring HF patients’ condition. To this end, the V-LAP platform can be utilized, since it incorporates the needed technological framework for transmitting multiple signals. With the advancements in technology and miniaturization, micro-devices will be further utilized to assess and treat various aspects of cardiac function in disease state and disease prevention.

## Figures and Tables

**Figure 1 micromachines-11-00902-f001:**
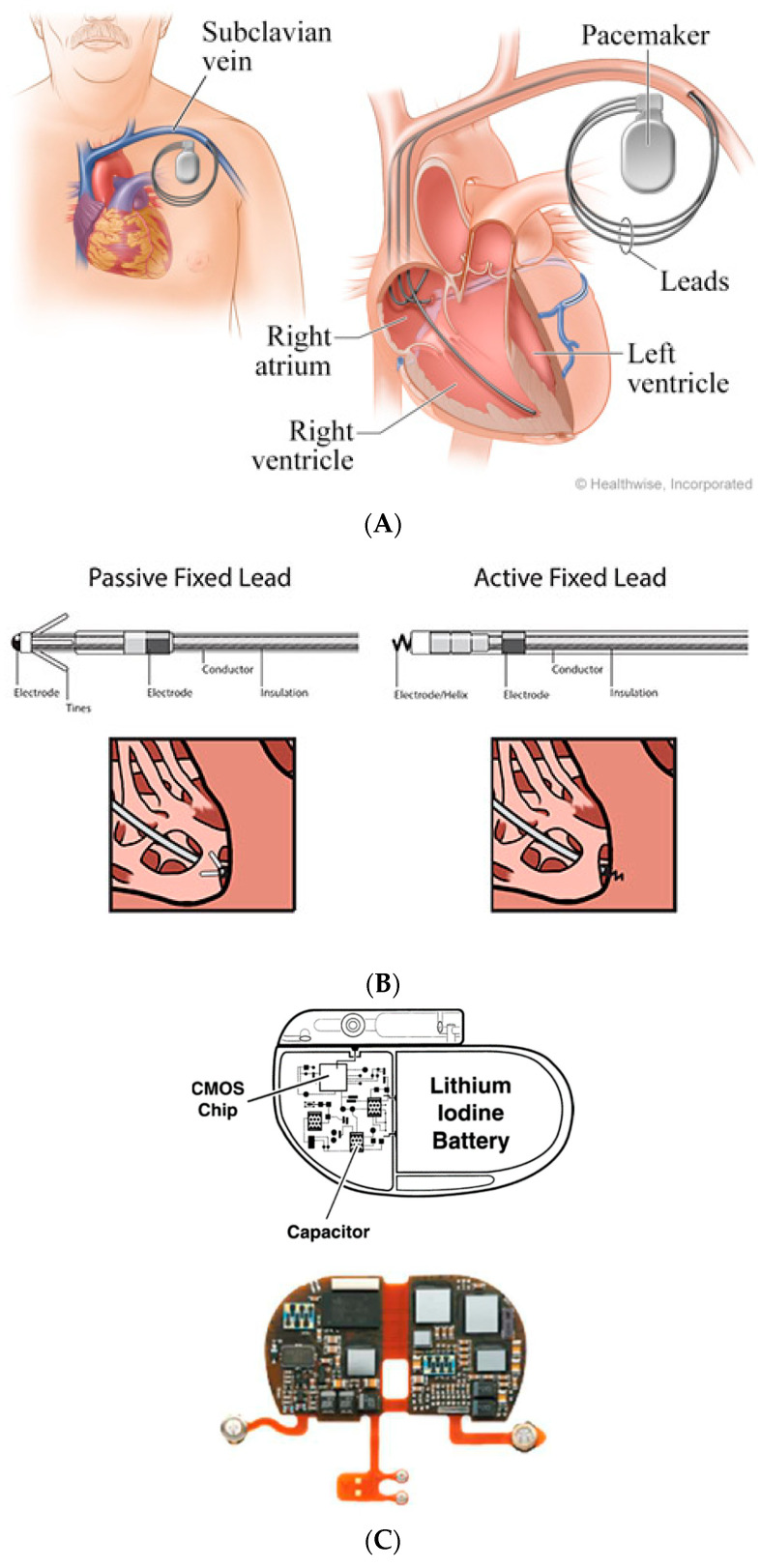
Schematic endovascular permanent pacemaker (PPM) system consisting of battery and programmable electrical chip assembly (together named ‘pacemaker’). (**A**) Special conductors (‘leads/electrodes’) implanted via a major central vein (endovascular approach) connect these pacemakers with the heart chambers (in this example, an atrial lead implanted within right atrium, ventricular lead implanted into right ventricle, and another lead implanted via the coronary sinus, enabling pacing of left ventricle). (**B**) Atrial and ventricular electrodes are fixed to the appropriate chambers by passive or active fixation mechanisms via tines or screws, respectively. (**C**) Pacemaker electrical chip assembly composed of large-scale integration circuits containing multiple complementary metallic oxide semiconductor (CMOS) chips integrated with resistors and capacitors.

**Figure 2 micromachines-11-00902-f002:**
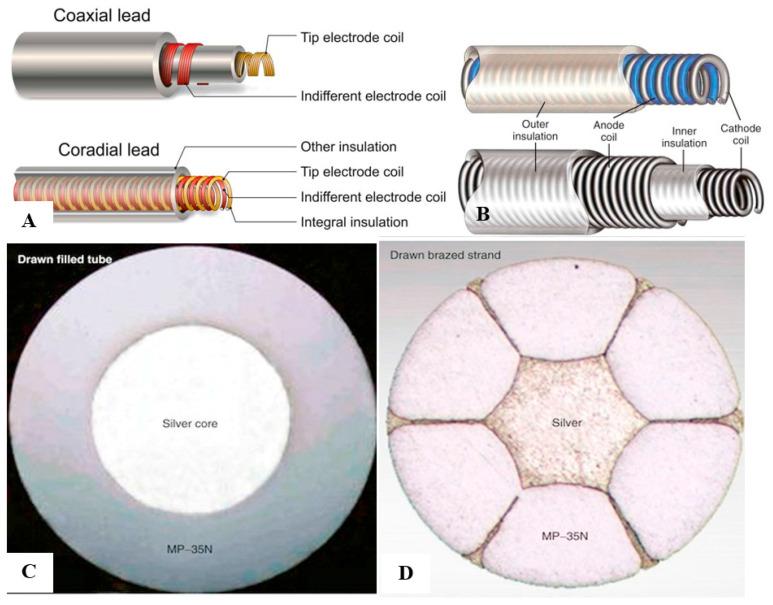
Pacemaker leads and their inner conductor design. Pacemaker leads conductors are typically of a coaxial or coradial designs, surrounded by insulations layers (**A**,**B**). Coaxial design conductors have an inner conductor that extends down the length of the lead to the tip with another outer coil conductor that also runs down the lead to the ring electrode, the anode. Coradial conductors are composed of a single coil conductor which extends down the length of the lead and consists of two parallel, alternating conductor strands, one of which connects to the cathode and the other to the anode. Each conductor strand is individually coated with an inner insulation layer, which serves to insulate each strand from the other, despite being intertwined. The single two-component coil is surrounded by another outer insulation layer. A magnified scheme of the inner and outer insulation layers of a typical coaxial lead are shown (**B**). The conductors themselves are usually made of silver core alloy wires, consisting of a central core of relatively soft, low-resistance silver surrounded by a thick strong high-resistance MP-35N outer shell in the drawn filled tube design (**C**), or by few strands of high-resistance MP-35N shells in the *drawn brazed strand* design (**D**).

**Figure 3 micromachines-11-00902-f003:**
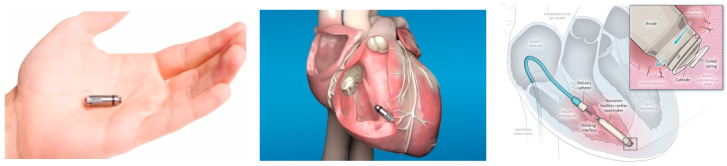
Novel miniature leadless pacemaker device (**left**) implanted directly into the right ventricle via a special delivery system (**middle**, **right**). There are no endovascular leads that are associated with this device. In this example, Micra leadless PPM is shown (Reproduced with permission of Medtronic, Inc.).

**Figure 4 micromachines-11-00902-f004:**
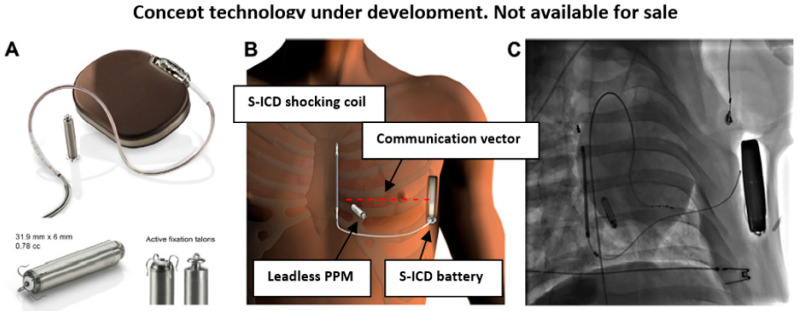
(**A**) Modular cardiac defibrillator and leadless pacemaker prototype. (**B**,**C**) Schematic depiction (**B**) and fluoroscopy image (**C**) of this human modular system, composed of subcutaneous ICD (EMBLEM MRI) with its long subcutaneous lead and shocking coil, and a miniaturized leadless PPM (EMPOWER™) positioned within right ventricle at some orientation angle to the radio-frequency communication vector (red--) (All photographs taken by Boston Scientific).

**Figure 5 micromachines-11-00902-f005:**
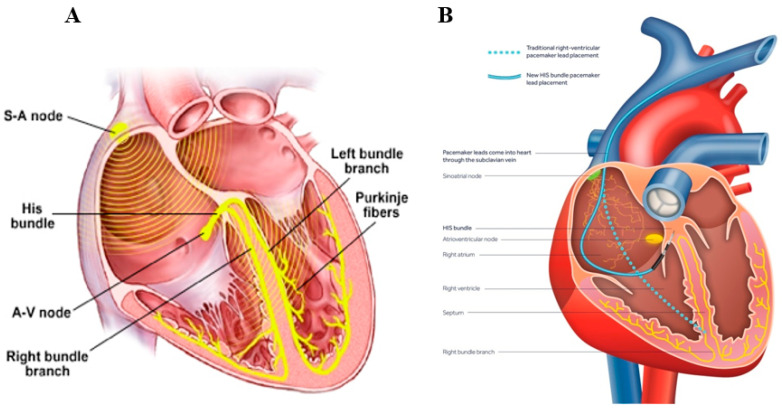
Novel ‘physiologic’ His pacing. (**A**) The physiologic conduction system, enabling synchronized propagation of the electrical impulse generated by the physiologic “pacemaker” called the sinoatrial node. This conduction system is composed of the His bundle, right and left bundle branches which are located along either side of the inter-ventricular septum, spreading into millions of tiny conductive Purkinje fibers, enabling a synchronized impulse propagation throughout the cardiac cells to result in a synchronized and efficient cardiac mechanical contraction. (**B**) Comparing the distal location of a conventional right ventricle (RV) lead (---) to a novel His pacing lead (**^__^**). Pacing via conventional RV lead which is positioned close to RV apex (outside the conduction system) causes a desynchronized conduction, while His pacing utilizes the physiological conduction system to ideally synchronize all cardiac cells.

**Figure 6 micromachines-11-00902-f006:**
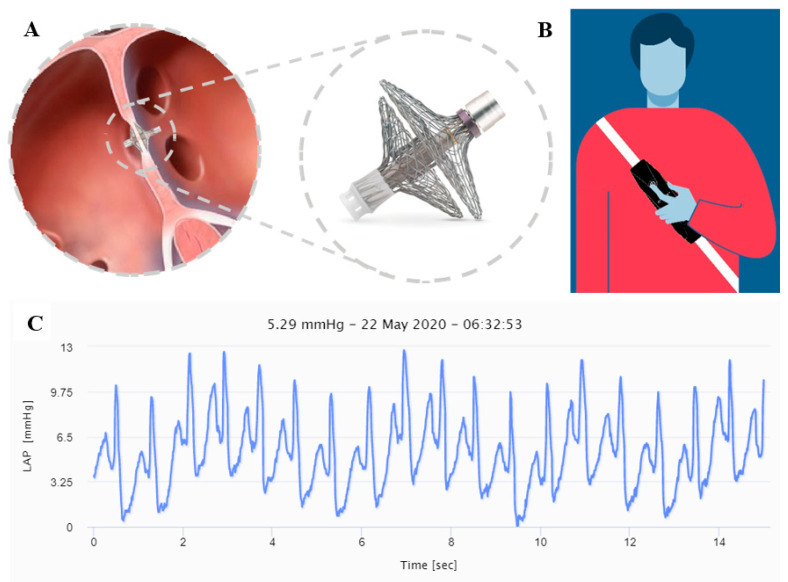
V-LAP heart failure (HF) monitoring system composed of a miniature cardiac sensor (**A**), and a portable external wearable device (**B**). The system enables a continuous high-resolution left atrial pressure (LAP) waveform recording (**C**).

**Table 1 micromachines-11-00902-t001:** Comparison between the two clinical leadless permanent pacemakers (PPM) systems.

Parameter	Micra (Medtronic, Inc., USA)	Nanostim (Saint Jude Medical, USA)
Volume	0.8 cm^3^	1 cm^3^
Size	25.9 mm × 6.7 mm	42 mm × 5.99 mm
Weight	1.75 g	2 g
Fixation	Passive nitinol tines	Active helix screw
Battery	lithium silver oxide/carbon monofluoride	lithium carbon monofluoride
Estimated battery longevity	10 years	14.7 years
Rate response mechanism	3-D accelerometer	Blood temperature sensing
Communication with remote programmer-‘telemetry’	Radio-frequency signals	Conductive communication through skin electrodes

**Table 2 micromachines-11-00902-t002:** Comparison between implantable heart failure monitoring devices.

Parameter	Chronicle	CardioMEMS	HeartPOD	V-LAP
Site of pressure readout	Right ventricle	Pulmonary artery	Left atrium	Left atrium
Structure	Subcutaneous device with intravenous lead in right ventricle	Implant and external antenna	subcutaneous antenna coil with intravenous trans-septal lead	Implanted within interatrial septum
Energy source	Subcutaneous battery	External	External	External
Clinical study	COMPASS-HF [33]	CHAMPION [34]	LAPTOP-HF [35]	Ongoing
Key findings	Failed in primary outcome	Reduced HF hospitalizations	Stopped early for implant-related complications	Pending
Current status	Not available for clinical use	FDA approved	Not available for clinical use	On trial

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
