# Peer review of "Cardiac Implantable Electronic Miniaturized and Micro Devices"

_micromachines, 2020, doi:10.3390/mi11100902_

Round 1

Reviewer 1 Report

The manuscript is improved. The minor comments below are mostly related to syntax. The major comments are aimed at improving the manuscript’s clinical accuracy.

Minor Comments:

1. On lines 13-14, change: “These include pacing devices to prevent low heart rates or terminate heart rhythm abnormalities ('arrhythmias')…” to “These include pacing devices to prevent low heart rates or terminate rapid heart rhythm abnormalities ('tachyarrhythmias')…”.
2. Wherever applicable change: “PPM’s” to “PPMs”. Likewise, CRT’s to CRTs and ICD’s to ICDs.
3. On lines 25-26, delete: “of these leads”.
4. On line 110, change: “spreading infectious” to “potentially spreading infectious”.
5. On line 135, change: “are usually engulfed” to “may become encapsulated”.
6. On line 148, change: “altitude” to “magnitude”.
7. Under Modular Subcutaneous ICD and leadless PPM consider mentioning the name EMPOWER™ leadless cardiac pacemaker (LCP) and noting that it is an investigational device from Boston Scientific. This is optional since it is noted in
8. On line 235, delete: “fast” which is functionally redundant.
9. On line 283, delete “subcutaneous” it’s already noted in line 282.
10. On line 345, change: “lower limb embolus” to “systemic embolism”.
11. On lines 352, 354, 358 and 365: His is a name (Wilhelm His Jr.) and should be capitalized. This also needs correction in figure 5.
12. On line 368, change: “conductive” to “conduction”.
13. On lines 389-420, all hyphenated words, except HF-related, can be combined into single words without hyphens. Likewise, trans-venously (line 439) should not be hyphenated.
14. On lines 415-416, change: “a part” to “apart”.
15. On line 454, change: “web-application” to “web application”.
16. On line 465, change: “intra-cardiac” to “intracardiac”.

Major Comments:

1. On lines 20-21, change: “… and cardiac re-synchronization devices that improve cardiac function and survival of HF patients to “and cardiac resynchronization devices that may improve cardiac function and survival in appropriately selected rom subtleHF patients”. This change tacitly acknowledges that a significant proportion of HF patients are non-responders.
2. A similar change should be made on lines 270-272.
3. Likewise on lines 50-51, change: “…were shown to attenuate HF symptoms and improve survival of HF patients” to “...were shown to attenuate HF symptoms and may improve survival in appropriately selected HF patients”.
4. On line 43, change: “via high-energy shock” to “via anti-tachycardia pacing or shocks of up to 42 Joules (BIOTRONIK, Berlin Germany)”. Definitions of high and low energy shocks may vary (see: Prehosp Emerg Care. 2010 Jan-Mar; 14(1): 62–70). For the same reason, delete “high- energy” from line 239.
5. Avoid the use of the abbreviation DFT which, in ICDs, refers to defibrillation threshold. For consistency you may wish to avoid DBS as well.
6. On lines 107-108 re: Regarding lead dislodgement [note spelling], there are early displacements, which occur within the first six weeks after implantation, and late displacements, after this period of time (see: Indian Pacing Electrophysiol J. 2003 Oct-Dec; 3(4): 231–238).
7. On line 244, note that reduction in ICD size has primarily depended on reduction in capacitor size.
8. On line 274, change: “the coronary sinus” to “a cardiac venous branch of the coronary sinus”.
9. On lines 325-326: do you mean inappropriate pacing with potentially deleterious consequences?
10. On line 370, add: (related to atrial flutter and fibrillation) after the word “strokes”.
11. On line 515 of figure 1 change: “into” to “via the”.
12. In figure 1, most commercially available cardiac venous leads do not have fixation mechanisms. Please amend the figure legend accordingly.

Reviewer 2 Report

In this paper, the authors reviewed implantable miniaturized devices and the basic algorithms and technologies underlying their development. The manuscript is properly organized and it provides a thorough summary of results in the literature including permanent pacemakers (PPM), defibrillators capable and cardiac re-synchronization devices for heart failure (HF) treatment applications. Overall, it is an valuable manuscript and can provide helpful information to researchers interested in cardiac implantable electronics. The reviewer recommends the publication of the paper after the following issues are addressed.

  1. It is better to highlight the importance of some referenced works and give some insightful comments about them, which can facilitate readers' understandings.
  2. What’s the impact of cardiac implantable electronic devices on myocardial contraction? How to characterize the influence?
  3. It is noticed that most introduced devices in the manuscript are millimeters in size. So, how should we define the “micro devices”? What role do implantable micro devices play in heart-related biomedical applications?
  4. In line 464, page 13, it’s mentioned that leadless CRT pacing systems are with high complication rates, is there any strategy to overcome the challenge at present?
  5. The picture letter numbers (a, b, c ...) in the figures are not uniform. For example, letter number in figure 4 and figure 6 are written in capitals (A, B, C), while figure 6 are written in lower-case (a, b, c).
  6. This review contains imperfect utilizations of abbreviation of technical terms. For example, the phrase “Heart failure (HF)” appears for 3 times (line 15, line 470, line 381), “permanent pacemakers (PPM)” (line 19, line 39) vs. “pacemaker (PPM)” (line 401).

Round 2

Reviewer 1 Report

The manuscript is vastly improved. Attention to the following will complete it. 

Minor comments:

  1. Lines 48-49: The change: “…were shown to attenuate HF symptoms and improve survival of HF patients” to “...were shown to attenuate HF symptoms and may improve survival in appropriately selected HF patients” was not properly completed. To simplify, please write “...were shown to attenuate HF symptoms and may improve survival in HF patients”.
  2. Line 303: cardiomyocytes is misspelled once as cariomyocytes.

Major comments:

  1. Line 40: Rather than choose an arbitrary definition, delete “in the range of 30-42 Joules” and change the parentheses to (“direct current cardioversion/defibrillation”).
  2. Line 224: Likewise, delete “30-42 J”.
  3. Lines 47-48: Delete (these devices however do not impact the cardiac contraction at the cellular level). The process of reverse remodeling may be influenced by changes at the cellular level.
  4. Lines 352-353: The sentence “Nevertheless, to the best of our knowledge, miniaturized devices for His or LBBB pacing are not currently available” is confusing. Do you mean “Commercially available pulse generators are compatible with conduction system pacing”? Or do you mean “Miniaturized devices designed specifically for His or LBBB pacing are not currently available”? Either change would suffice or the current sentence could simply be eliminated.
  5. I agree that RA and RV leads have fixation mechanisms. Therefore, the word “Most” can be deleted from the 1B figure legend. To clarify my comments, the terms cardiac venous or cardiac veins refer to the heart’s venous system (not the great veins of the thorax or the cardiac chambers). Cardiac veins are not necessarily anatomically associated with the coronary arteries. Hence, the term coronary veins is a misnomer.

Author Response

The manuscript is vastly improved. Attention to the following will complete it. 

Minor comments:

  1. Lines 48-49: The change: “…were shown to attenuate HF symptoms and improve survival of HF patients” to “...were shown to attenuate HF symptoms and may improve survival in appropriately selected HF patients” was not properly completed. To simplify, please write “...were shown to attenuate HF symptoms and may improve survival in HF patients”.

Answer: The text was changed according to the reviewer suggestion.

  1. Line 303: cardiomyocytes is misspelled once as cariomyocytes.

Answer: Indeed, misspelling was corrected.

Major comments:

  1. Line 40: Rather than choose an arbitrary definition, delete “in the range of 30-42 Joules” and change the parentheses to (“direct current cardioversion/defibrillation”).

Answer: Change done.

  1. Line 224: Likewise, delete “30-42 J”.

Answer: Change done.

  1. Lines 47-48: Delete (these devices however do not impact the cardiac contraction at the cellular level). The process of reverse remodeling may be influenced by changes at the cellular level.

Answer: I accept the reviewer right comment and deleted these lines.

  1. Lines 352-353: The sentence “Nevertheless, to the best of our knowledge, miniaturized devices for His or LBBB pacing are not currently available” is confusing. Do you mean “Commercially available pulse generators are compatible with conduction system pacing”? Or do you mean “Miniaturized devices designed specifically for His or LBBB pacing are not currently available”? Either change would suffice or the current sentence could simply be eliminated.

Answer: Indeed, we wanted to say that "Miniaturized devices designed specifically for His or LBBB pacing are not currently available" and the text was changed accordingly.

  1. I agree that RA and RV leads have fixation mechanisms. Therefore, the word “Most” can be deleted from the 1B figure legend. To clarify my comments, the terms cardiac venous or cardiac veins refer to the heart’s venous system (not the great veins of the thorax or the cardiac chambers). Cardiac veins are not necessarily anatomically associated with the coronary arteries. Hence, the term coronary veins is a misnomer.

Answer: In accordance with the reviewer comment the word "most" deleted from fig 1 legend.

This manuscript is a resubmission of an earlier submission. The following is a list of the peer review reports and author responses from that submission.

Round 1

Reviewer 1 Report

This is an interesting manuscript. Attention to the following should improve it.

Minor comments:

  1. While generally good, attention to improving the English syntax would be useful.

Major comments:

  1. Under Multi-component leadless cardiac resynchronization pacing system: Note that cardiac resynchronization may also be achieve by His bundle or left bundle branch pacing.
  2. You may wish to comment on the battery and docking button issues related to Nanostim.
  3. It is important to note that any device implanted in the left atrium of left ventricular endocardium may present a risk for thrombus formation and thromboembolism (in particular stroke).
  4. Under Implantable long-term electrocardiographic monitoring device, the sentence “Once arrhythmia is detected, it is recorded by the ILR which than sends an automatic alarm to the patient's clinic via internet connection, to facilitate appropriate therapy” is potentially misleading. Change it to: “Once arrhythmia is detected, it is recorded by the ILR which than sends an automatic alarm to the patient's clinic via internet connection, to facilitate diagnosis and selection of additional appropriate therapy”.
  5. Consider deleting lines 258-266 as the Chronicle had little clinical impact and is currently irrelevant.

Reviewer 2 Report

The authors provide a review of various recent miniaturized implantable cardiac electronic devices. Although there is a discussion on MEMS-based implantable pressure sensor by CardioMEMS, most of the manuscript focuses on the development of leadless implantable pacing devices, which are definitely smaller than conventional PMs or ICDs. In general, this review would be better served for a more clinical audience in cardiology rather than that of Micromachines, which focuses on "micro/nano-scaled structures, materials, devices and systems".  All of the examples certainly have some IC components but there is no discussions on novel microscale sensors, electrodes, materials, etc. Furthermore, there are a lot of emphasis on the results of clinical trials on these miniaturized implantable electronic systems but little discussions on physics and mechanisms of functionality for these implants that the audience would find more interesting. The authors also make several claims/conjectures that are not fully supported by primary literature citations. Finally, the manuscript is very poorly written in terms of both grammar and style. There are countless run-on sentences that span 10's of lines and a number of informal use of language. 

Specific comments.

  • Title: None of the systems described are actually micro devices. 
  • No formal introduction section?
  • Line 29: 10-line-long single sentence?? Inappropriate combination of quotation marks, paratheses, spacing, signs (+,&) and acronyms. 
  • Line 45: these leads are silver-cored alloy wires. 
  • line 93: Pacing requires relatively small amount of energy, which is why these leadless pacing is possible. And the same reason is why there is no examples of miniaturized ICDs because it requires larger current delivery.
  • Line 94: Impedance due to wires are negligible compare to the impedance due to the electrode and body interface
  • Line 122: How are these no longer "foreign materials"? Why is this in quotation? This is at best conjecture. Probably due to overall smaller surface area
  • Line 191-223: One reference for this entire section??
  • Line 214: should be micro (\mu) W, not uW. 
  • Implantable devices t monitor heart failure
    • This is a section that could highlight development of implantable microscale sensing technologies but most of this discussion is very generic with focus on clinical research results rather than the details of sensors physics, packaging, and other aspects of the microdevice development.
  • Conclusions: there is no separate section. And there is a lack of critical discussion on advances and opportunities for future implantable microscale electronic devices for cardiac applications.
  • Figures: Small text are not visible and in some cases copyright logos are still visible. The authors should pay more attention to the development of these figures than to just copy and paste from other works. Fig 3 has "Figure 1" still written on top from its original journal. 
  • Author contribution, funding, acknowledgement sections not filled out
  • One of the authors clearly has a conflict of interest for the V-LAP devices but is not indicated here.